# On the surprising similarities between supervised and self-supervised models

**Robert Geirhos**[§]
University of Tübingen & IMPRS-IS

**Kantharaju Narayanappa**
University of Tübingen

**Benjamin Mitzkus**
University of Tübingen

**Matthias Bethge**[*]
University of Tübingen

**Felix A. Wichmann**[*]
University of Tübingen

**Wieland Brendel**[*]
University of Tübingen

[*]Joint senior authors
[§]robert.geirhos@uni-tuebingen.de

## Abstract

How do humans learn to acquire a powerful, flexible and robust representation of objects? While much of this process remains unknown, it is clear that humans do not require millions of object labels. Excitingly, recent algorithmic advancements in self-supervised learning now enable convolutional neural networks (CNNs) to learn useful visual object representations without supervised labels, too. In the light of this recent breakthrough, we here compare self-supervised networks to supervised models and human behaviour.

We tested models on 15 generalisation datasets for which large-scale human behavioural data is available (130K highly controlled psychophysical trials). Surprisingly, current self-supervised CNNs share four key characteristics of their supervised counterparts: (1.) relatively poor noise robustness (with the notable exception of SimCLR), (2.) non-human category-level error patterns, (3.) non-human image-level error patterns (yet high similarity to supervised model errors) and (4.) a bias towards texture. Taken together, these results suggest that the strategies learned through today's supervised and self-supervised training objectives end up being surprisingly similar, but distant from human-like behaviour. That being said, we are clearly just at the beginning of what could be called a self-supervised revolution of machine vision, and we are hopeful that future self-supervised models behave differently from supervised ones, and—perhaps—more similar to robust human object recognition.

## 1 Introduction

*"If intelligence is a cake, the bulk of the cake is unsupervised learning, the icing on the cake is supervised learning and the cherry on the cake is reinforcement learning"*, Yann LeCun famously said [1]. Four years later, the entire cake is finally on the table—the representations learned via self-supervised learning now compete with supervised methods on ImageNet [2] and outperform supervised pre-training for object detection [3]. But given this fundamentally different learning mechanism, *how do recent self-supervised models differ from their supervised counterparts in terms of their behaviour?*

We here attempt to shed light on this question by comparing eight flavours of "cake" (PIRL, MoCo, MoCoV2, InfoMin, InsDis, SimCLR-x1, SimCLR-x2, SimCLR-x4) with 24 common variants of

"icing" (from the `AlexNet`, `VGG`, `Squeezenet`, `DenseNet`, `Inception`, `ResNet`, `ShuffleNet`, `MobileNet`, `ResNeXt`, `WideResNet` and `MNASNet` star cuisines). Specifically, our culinary test buffet aims to investigate:

1. Are self-supervised models more robust towards distortions?

2. Do self-supervised models make similar errors as either humans or supervised models?

3. Do self-supervised models recognise objects by texture or shape?

For all of these questions, we compare supervised and self-supervised[1] models against a comprehensive set of openly available human psychophysical data totalling over 130,000 trials [4, 5]. This is motivated on one hand by the fact that humans, too, rapidly learn to recognise new objects without requiring hundreds of labels per instance; and on the other hand by a number of fascinating studies reporting increased similarities between self-supervised models and human perception. For instance, Lotter et al. [6] train a model for self-supervised next frame prediction on videos, which leads to phenomena known from human vision, including perceptual illusions. Orhan et al. [7] train a self-supervised model on infant video data, finding good categorisation accuracies on some (small) datasets. Furthermore, Konkle and Alvarez [8] and Zhuang et al. [9] report an improved match with neural data; Zhuang et al. [9] also find more human-like error patterns for semi-supervised models and Storrs and Fleming [10] observe that a self-supervised network accounts for behavioural patterns of human gloss perception. While methods and models differ substantially across these studies, they jointly provide evidence for the intriguing hypothesis that self-supervised machine learning models may better approximate human vision.

## 2   Methods

**Models.** `InsDis` [11], `MoCo` [12], `MoCoV2` [13], `PIRL` [3] and `InfoMin` [14] were obtained as pre-trained models from the PyContrast model zoo. We trained one linear classifier per model on top of the self-supervised representation. A PyTorch [15] implementation of `SimCLR` [2] was obtained via simclr-converter. All self-supervised models use a `ResNet-50` architecture and a different training approach within the framework of contrastive learning [e.g. 16]. For supervised models, we used all 24 available pre-trained models from the PyTorch model zoo version 1.4.0 (`VGG`: with batch norm).

**Linear classifier training procedure.** The PyContrast repository by Yonglong Tian contains a Pytorch implementation of unsupervised representation learning methods, including pre-trained representation weights. The repository provides training and evaluation pipelines, but it supports only multi-node distributed training and does not (currently) provide weights for the classifier. We have used the repository's linear classifier evaluation pipeline to train classifiers for InsDis [11], MoCo [12], MoCoV2 [13], PIRL [3] and InfoMin [14] on ImageNet. Pre-trained weights of the model representations (without classifier) were taken from the provided Dropbox link and we then ran the training pipeline on a NVIDIA TESLA P100 using the default parameters configured in the pipeline. Detailed documentation about running the pipeline and parameters can be found in the PyContrast repository (commit #3541b82).

**Datasets.** Models were tested on 12 different image degradations from [4], as well as on texture-vs-shape datasets from [5]. Plotting conventions follow these papers (unless indicated otherwise).

## 3   Results

We here investigate four behavioural characteristics of self-supervised networks, comparing them to their supervised counterparts on the one hand and to human observers on the other hand: out-of-distribution generalisation (3.1), category-level error patterns (3.2), image-level error patterns (3.3), and texture/shape biases (3.4).

---

[1]"Unsupervised learning" and "self-supervised learning" are sometimes used interchangeably. We use the term "self-supervised learning"' since the methods use (label-free) supervision.

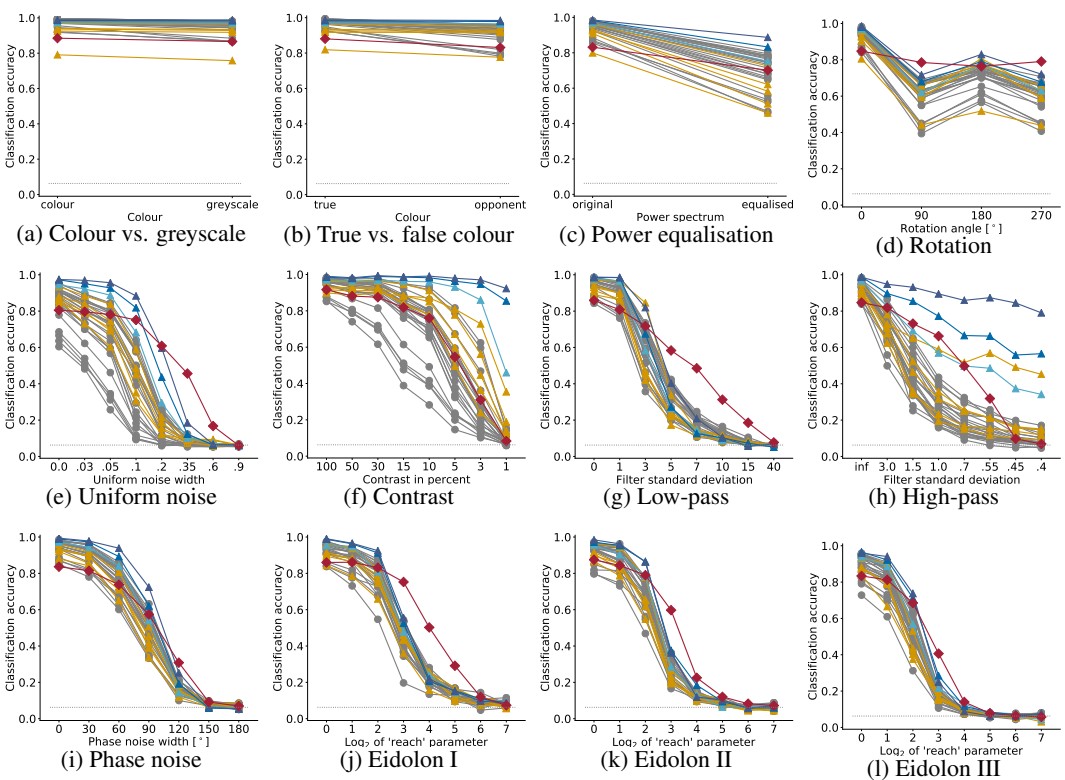

Figure 1: Noise generalisation results for humans (red diamonds) vs. supervised models (grey cicles) vs. self-supervised models (orange triangles). Self-supervised SimCLR variants: blue triangles.

## 3.1 With the exception of SimCLR, supervised and self-supervised models show similar (non-human) out-of-distribution generalisation

**Motivation.** Given sufficient quantities of labelled training data, CNNs can learn to identify objects when the input images are noisy. However, supervised CNNs typically generalise poorly to novel distortion types not seen during training, so-called out-of-distribution images [4]. In contrast, human perception is remarkably robust when dealing with previously unseen types of noise. Given that recent self-supervised networks are trained to identify objects under a variety of transformations (like scaling, cropping and colour shifts), have they learned a more robust, human-like representation of objects, where high-level semantic content is unimpaired by low-level noise?

**Results.** In Figure 1, we compare self-supervised and supervised networks on twelve different types of image distortions. Human observers were tested on the exact same distortions by [4]. Across distortion types, self-supervised networks are well within the range of their poorly generalising supervised counterparts. However, there is one exception: `SimCLR` shows strong generalisation improvements on uniform noise, low contrast and high-pass images—quite remarkable given that the network was trained using other augmentations (random crop with flip and resize, colour distortion, and Gaussian blur). Apart from `SimCLR`, however, we do not find benefits of self-supervised training for distortion robustness. These results for ImageNet models contrast with [17] who observed some robustness improvements for a self-supervised model trained on the CIFAR-10 dataset.

## 3.2 Self-supervised models make non-human category-level errors

**Motivation.** On clean images, CNNs now recognise objects as well as humans. But do they also confuse similar categories with each other (which can be investigated using confusion matrices)?

**Results.** In Figure 4 (moved to appendix for space reasons), we compare category-level errors of humans against a standard supervised CNN (`ResNet-50`) and three self-supervised CNNs. We chose

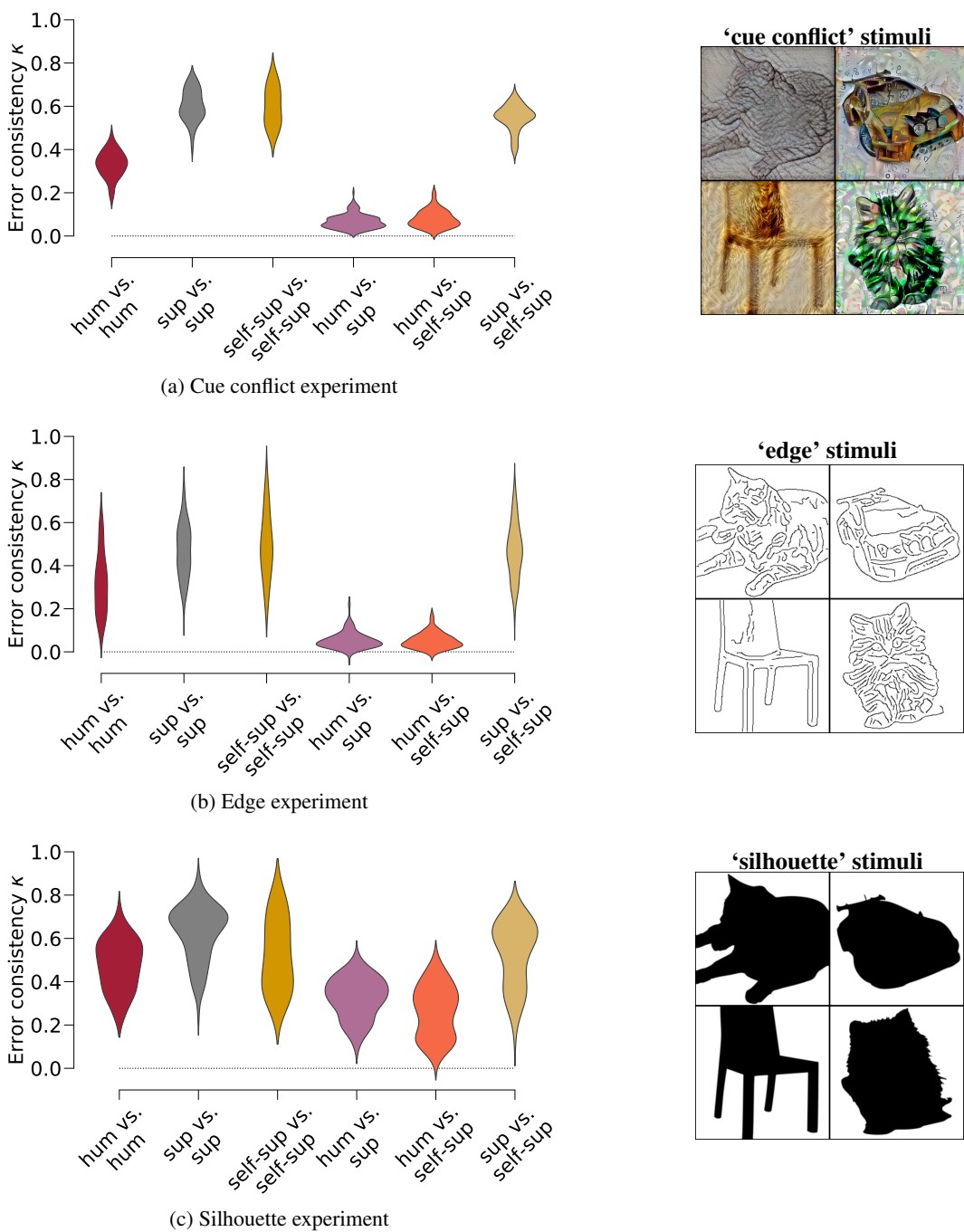

(a) Cue conflict experiment

(b) Edge experiment

(c) Silhouette experiment

Figure 2: Self-supervised models make errors on the same images as supervised models. Error consistency (high $\kappa$ = consistent errors) between all combinations of the following three groups: humans (hum), supervised networks (sup) and self-supervised networks (self-sup). Stimuli from [5]: (a) cue conflict, (b) edges and (c) silhouettes (visualisation by [18]). For all three experiments, consistency between networks is much higher than between networks and humans: CNNs make errors on the same images as other CNNs, whether these are supervised or self-supervised.

uniform noise for this comparison since this is one of the noise types where `SimCLR` shows strong improvements, nearing human-level accuracies. Looking at the confusion matrices, both humans and CNNs start with a dominant diagonal indicating correct categorisation. With increasing noise level, however, all CNNs develop a strong tendency to predict one category and only one, such as "knife" for `ResNet-50`. Human observers, on the other hand, more or less evenly distribute their errors across classes. For supervised networks, this pattern of errors was already observed by [4]; we here find that this peculiar idiosyncrasy is shared by self-supervised networks, indicating that discriminative supervised training is not the underlying reason for this non-human behaviour.

### 3.3 Self-supervised models make non-human image-level errors (error consistency)

**Motivation.** Achieving human-level accuracies on a dataset does not necessarily imply using a human-like strategy: different strategies can lead to similar accuracies. Therefore, it is essential to investigate *on which stimuli* errors occur. If two decision makers—for instance, a human observer and a CNN—use a similar strategy, we can expect them to consistently make errors on the same individual images. This intuition is captured by *error consistency* ($\kappa$), a metric to measure the degree to which decision makers make the same image-level errors [18]. $\kappa > 0$ means that two decision makers systematically make errors on the same images; $\kappa = 0$ indicates no more error overlap than what could be expected by chance alone.

**Results.** Figure 2 plots the consistency of errors (measured by $\kappa$). Humans make highly similar errors as other humans (mean $\kappa = 0.32$), but neither supervised nor self-supervised models make human-like errors. Instead, error consistency *between* model groups (self-supervised vs. supervised) is just as high as consistency *within* model groups: self-supervised models make errors on the same images as supervised models, an indicator for highly similar strategies.

### 3.4 Self-supervised models are biased towards texture

**Motivation.** Standard supervised networks recognise objects by relying on local texture statistics, largely ignoring global object shape [5, 19, 20]. This striking difference to human visual perception has been attributed to the fact that texture is a shortcut sufficient to discriminate among objects [21]—but is texture also sufficient to solve self-supervised training objectives?

**Results.** We tested a broad range of CNNs on the texture-shape cue conflict dataset from [5]. This dataset consists of images where the shape belongs to one category (e.g. cat) and the texture belongs to a different category (e.g. elephant). When plotting whether CNNs prefer texture or shape (Figure 3), we observe that most self-supervised models have a strong texture bias known from traditional supervised models. This texture bias is less prominent for `SimCLR` (58.3–61.2% texture decisions), which is still on par with supervised model `Inception-V3` (60.7% texture decisions). These findings are in line with [22], who observed that the influence of training data augmentations on shape bias is stronger than the role of architecture or training objective. Neither supervised nor self-supervised models have the strong shape bias that is so characteristic for human observers, indicating fundamentally different decision making processes between humans and CNNs.

## 4 Discussion

Comparing self-supervised networks to supervised models and human observers, we here investigated four key behavioural characteristics: out-of-distribution generalisation, category-level error patterns, image-level error patterns, and texture bias. Overall, we find that self-supervised models resemble their supervised counterparts much more closely than what could have been expected given fundamentally different training objectives. While standard models are notoriously non-robust [4, 21, 23–25], `SimCLR` represents a notable exception in some of our experiments as it is less biased towards texture and much more robust towards some types of distortions. It is an open question whether these benefits arise from the specific set of data augmentations used during `SimCLR` model training.

Perhaps surprisingly, error consistency analysis suggests that the images on which supervised and self-supervised models make errors overlap strongly, much more than what could have been expected by chance alone. This provides evidence for similar processing mechanisms: It seems that switching label-based supervision for a contrastive learning scheme does not have a strong effect on the inductive

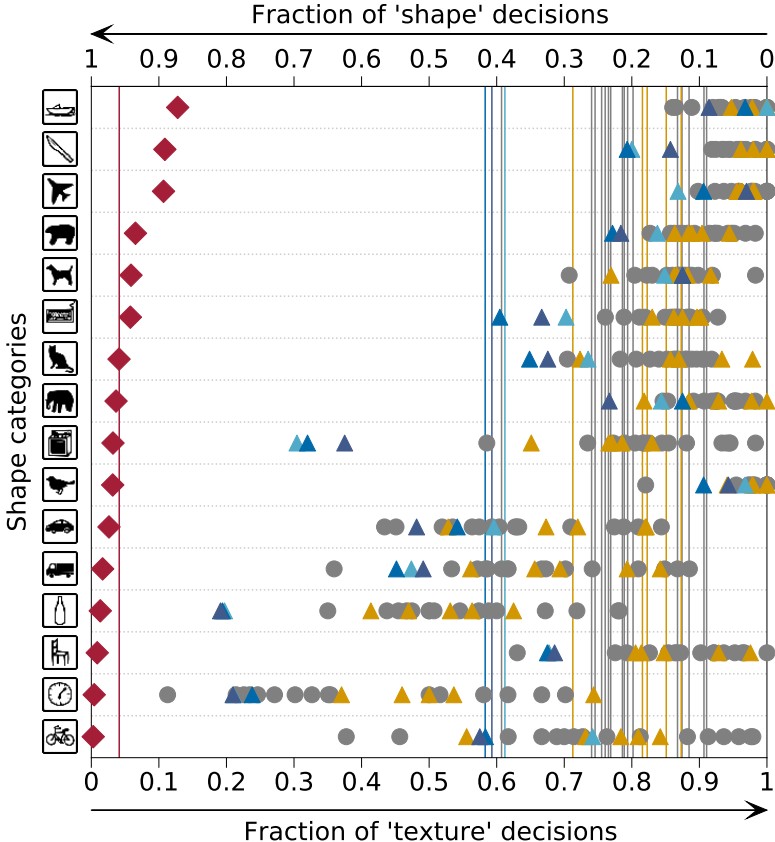

Figure 3: Self-supervised models are biased towards texture. Vertical lines indicate the average across categories for a certain model. Humans: red diamonds, supervised models: grey cicles, self-supervised models: orange triangles, self-supervised SimCLR variants: blue triangles.

bias of the resulting model—at least for the currently used contrastive approaches. Furthermore, we find little evidence for human-like behaviour in the investigated self-supervised models. While this investigation focused on state-of-the-art contrastive learning methods, other self-supervised methods might lead to different results.

We are clearly just witnessing the beginning of what could be called a self-supervised revolution of machine vision, and we expect future self-supervised models to behave significantly differently from supervised ones. What we are showing is that the current self-supervised CNNs are not yet more human-like in their strategies and internal representations than plain-vanilla supervised CNNs. We hope, however, that analyses like ours may facilitate the tracking of emerging similarities and differences, whether between different types of models or between models and human perception.

### Acknowledgement

We thank Santiago Cadena for sharing a PyTorch implementation of SimCLR, and Yonglong Tian for providing pre-trained self-supervised models on github. Furthermore, we are grateful to the International Max Planck Research School for Intelligent Systems (IMPRS-IS) for supporting R.G.; the Collaborative Research Center (Projektnummer 276693517—SFB 1233: Robust Vision) for supporting M.B. and F.A.W.; the German Federal Ministry of Education and Research through the Tübingen AI Center (FKZ 01IS18039A) for supporting W.B. and M.B.; and the German Research Foundation through the Cluster of Excellence "Machine Learning—New Perspectives for Science", EXC 2064/1, project number 390727645 for supporting F.A.W.

### Author contributions

Project idea: R.G. and W.B.; project lead: R.G.; implementing and training self-supervised models: K.N.; model evaluation pipeline: R.G., K.N. with input from W.B.; data visualisation: R.G. and B.M. with input from M.B., F.A.W. and W.B.; guidance, feedback, infrastructure & funding acquisition: M.B., F.A.W. and W.B.; paper writing: R.G. with input from all other authors.

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

## Appendix

Figure 4 shows confusion matrices for uniform noise, Figure 5 for low-pass filtering.

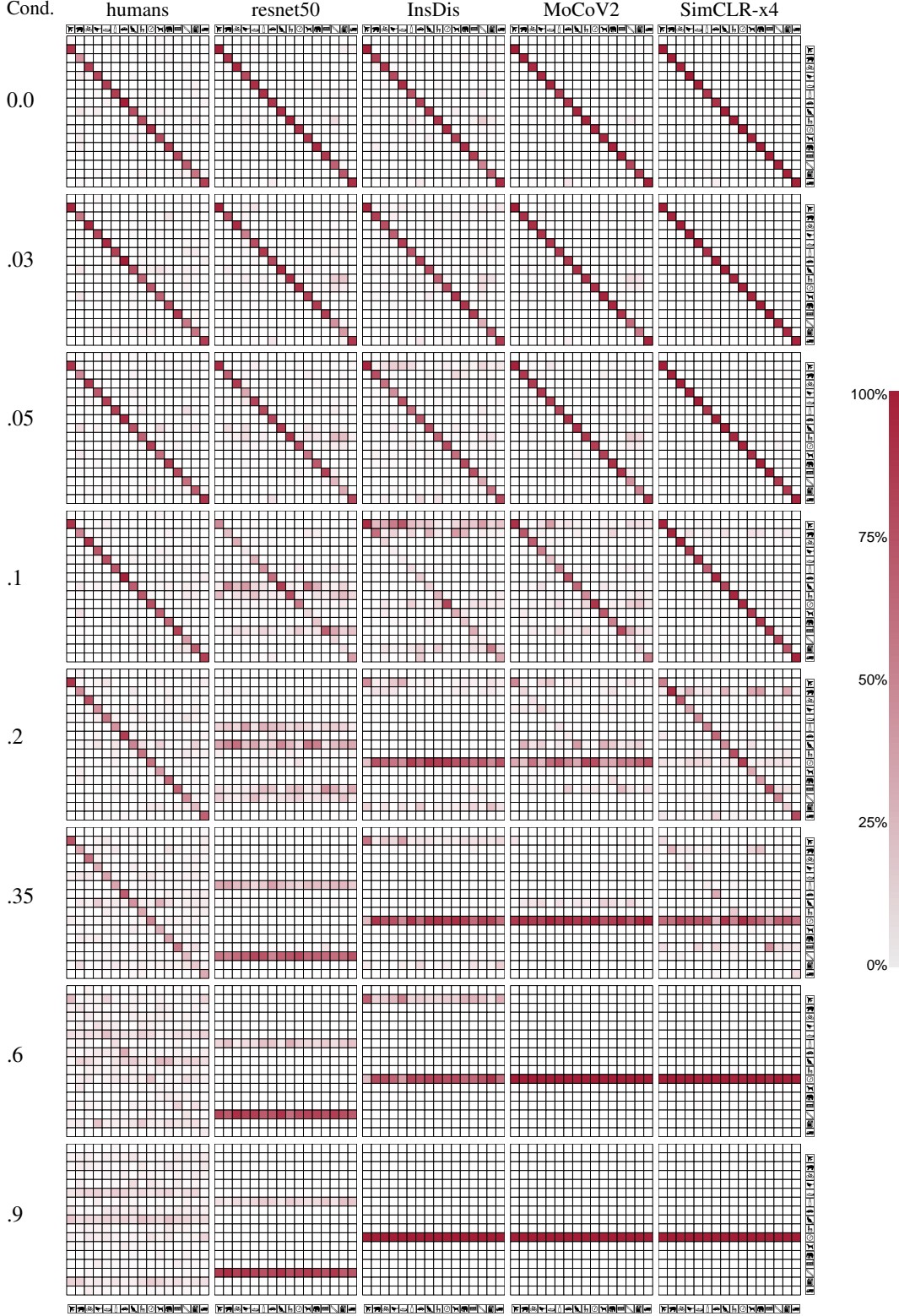

Figure 4: Confusion matrices for different conditions ("Cond.") of the uniform noise experiment. Columns show ground truth object categories, rows indicate predicted categories. Supervised `ResNet-50` and self-supervised networks `InsDis`, `MoCoV2` & `SimCLR-x4` all preferentially select a single category with increasing noise level.

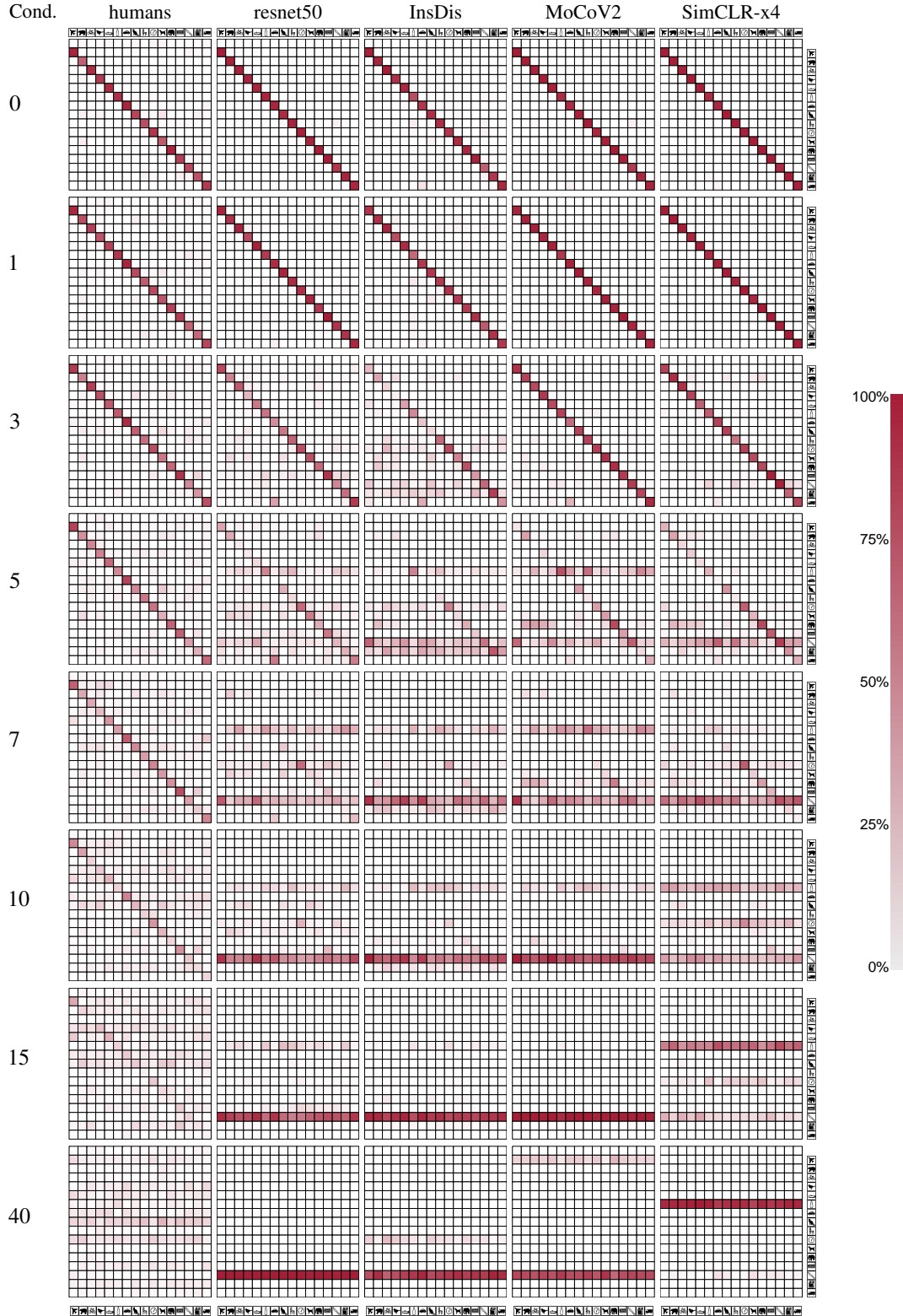

Figure 5: Confusion matrices for low-pass filtering. Again, CNNs develop a preference for a certain category as the distortion strength increases.

