# OpenReview forum: "On the surprising similarities between supervised and self-supervised models"
_NeurIPS.cc/2020/Workshop/SVRHM — SVRHM@NeurIPS Oral_

### Official Review · AnonReviewer2 · 2020-10-21
**paper review**

**Rating:** 9
**Confidence:** 4

**Review:**

Based on the recent successes of self-supervised methods for ImageNet pretraining, this paper asks the question of whether or not the predictions of the self-supervised models are more aligned with human behavior than fully supervised methods. The authors test a variety of supervised and self-supervised models in four different settings where they also have human experimental data: 1) out-of-distribution generalization, 2) category-level errors, 3) image-level errors, and 4) shape vs. texture bias. They find that self-supervised models largely make similar errors as supervised models, with the exception of SimCLR in the out-of-distribution generalization task. SimCLR is also slightly more shape biased than the other models, but nowhere near the level of humans.

Pros:
Thorough comparison in terms of number of models. The authors evaluated several models from both the supervised and unsupervised side
The paper is very clearly written and enjoyable to read
Nice set of chosen tasks paired with experimental human data

Cons:
In my opinion the only flaw is that it leaves me wanting to know more. Which elements really contribute to the slightly more human-like aspects of SimCLR? Is it data augmentations? The specific contrastive loss function used? Training with super large batch size? Would be great to see some follow up work on this topic

Overall I think this work is highly relevant to the SVRHM workshop as well as the greater machine learning community and is a strong accept.

---

### Official Review · AnonReviewer3 · 2020-10-26
**Compelling evidence that supervised and contrastive self-supervised models behave similarly**

**Rating:** 7
**Confidence:** 5

**Review:**

This submission investigates the performance of self-supervised models on out-of-distribution datasets, comparing their performance and predictions to supervised models and humans. It shows that self-supervised models generally behave similarly to supervised models and substantially different from humans.

Strengths:

The submission convincingly demonstrates its main point, that contrastive self-supervised learning does little to narrow the gap between human and machine perception. It is particularly surprising that self-supervised and supervised models tend to exhibit similar errors, as shown in Figure 2.

Experiments appear to be appropriately conducted and include many (24) different supervised models.

The writing is exceptionally clear and the paper is easy to follow.

Weaknesses:

The self-supervised models under investigation all involve slightly different flavours of contrastive learning. Although it is interesting to compare this class of models with supervised models, it is not entirely surprising that they are all relatively similar to each other. Given that contrastive self-supervised methods currently perform substantially better than alternative approaches, I think it’s fine to focus on these approaches, but particularly in the conclusion, I feel the paper could do more to acknowledge that these models reflect several flavours of a single self-supervised method and different results could potentially be observed even for existing models trained with other self-supervised methods.

It’s not clear whether the difference between humans and CNNs in Figure 3 reflects different underlying mechanisms or different evaluation procedures. If I understand correctly from ref. 4, humans saw many images in a single session. Assuming intact memory and metacognition, it seems unlikely that humans would provides the same prediction for all images where they could not confidently determine the class. The CNNs have no memory of previous images, which may explain why they do not attempt to distribute probability mass across classes. Looking closely at Figure 3, it is also clear that humans tend to pick some classes substantially more often than others in the highest noise condition.

L148 suggests the improved noise robustness of SimCLR could arise from the training objective. Since several of the models here use exactly the same training objective as SimCLR, the difference clearly does not arise from the training objective alone. SimCLR and MoCo v2 are especially similar, and except for momentum contrast itself, the differences (colour distortion strength, batch size, and optimizer) are equally relevant to supervised learning. Perhaps a better way of framing this point is that it is unclear whether the relevant design choices _interact_ with the training objective or whether similar effects would be observed if these design choices were applied to a supervised model. It would clearly strengthen the work if this experiment were performed.

Minor:

Figure 2 appears before Figure 3.

SimCLR’s colour augmentation actually includes random changes in contrast.

---

### Official Review · AnonReviewer1 · 2020-10-29
**Interesting Analysis**

**Rating:** 7
**Confidence:** 5

**Review:**

The paper performs an interesting quantitative analysis of the similarities/dis-similarities between supervised models, self-supervised models and humans.

Some interesting inferences and comments about them:
1) Self-supervised and supervised methods demonstrate poor generalization in the presence of noise, except SimCLR. Infact SimCLR demonstrates higher generalization than humans on some specific kinds of noise.
This result is really interesting. It would help to provide a justification/hypothesis for why this happens (perhaps because SimCLR has these forms of augmentation - blue, sobel filters, etc). It would be interesting to see some examples where humans fail and SimCLR succeeds.

2) Human errors are different from self-supervised /supervised models' errors
This is somewhat expected, but nevertheless good to have quantification.

3) Image-level Consistency among humans/supervised/self-supervised models
This result showing that there is consistency within groups but not with humans is also expected.
I think the fact that supervised and self-supervised models are image-level consistent is pretty interesting. It would've helped to have a quick explanation of the error consistency metric from [17]

4) Texture-bias of self-supervised models
I think this results reveals a previously unknown fact about self-supervised models. It is interesting to see that SimCLR overcomes these issues of texture bias simply by changing the augmentations (which is in line with previous findings in the supervised domain).

Overall I think this paper provides an interesting analysis and will serve as a good reference for the insights provided.

Some minor edits needed:
1) Rescale y-axis of row 1 of figure 1 to make it cleaner and more parsable.
2) Add a legend to Figure 4 (I assumed it to be the same as Fig 1).

---

### Public Comment · ~Robert_Geirhos1 · 2020-11-24
**Author's update on camera-ready version**

First of all, we would like to thank all three reviewers for their valuable feedback and their assessment of our paper/results as "really interesting" (R1), "highly relevant to the SVRHM workshop as well as the greater machine learning community" (R2) and "compelling" & "exceptionally clear" (R3).

While we are actively working on extending the presented analysis; we were able to incorporate some important reviewer suggestions for the camera ready version already - e.g. we added a legend/model description to the shape bias plot, stated more clearly in the discussion that our experiments investigated one particular highly successful method of self-supervised learning (contrastive approaches), re-phrased a statement which initially suggested that the training objective alone may be responsible for the outstanding performance of SimCLR, and we changed the colour scheme and plotting symbols to make our figures colourblind-friendly and readable when printed in grayscale.

---

### Decision · Program_Chairs · 2020-11-02

Accept (Oral)